# How to Identify Potentials and Barriers of Raw Materials Recovery from Tailings? Part I: A UNFC-Compliant Screening Approach for Site Selection

Rudolf Suppes [1,2,*] and Soraya Heuss-Aßbichler [3]

1    Institute of Mineral Resources Engineering (MRE), RWTH Aachen University, Wüllnerstr. 2,
     52062 Aachen, Germany
2    CBM GmbH—Gesellschaft für Consulting, Business und Management mbH, Horngasse 3,
     52064 Aachen, Germany
3    Department of Earth and Environmental Sciences, Ludwig-Maximilians-Universität München,
     Theresienstr. 41, 80333 Munich, Germany; soraya@min.uni-muenchen.de
*    Correspondence: suppes@cbm-ac.de

**Abstract:** Mapping the raw material (RM) potential of anthropogenic RMs, such as tailings, requires a comprehensive assessment and classification. However, a simple procedure to quickly screen for potentially viable RMs recovery projects similar to reconnaissance exploration of natural mineral RMs is missing. In this article, a quick and efficient approach to systematically screen tailings storage facilities (TSFs) is presented to evaluate if a particular TSF meets the criteria to be assessed in a more advanced study including costly on-site exploration. Based on aspects related to a TSF's contents, physical structure, surroundings, potential environmental and social impacts, and potentially affected stakeholders, it guides its user in compiling the information at local scale in a structured manner compliant with the United Nations Framework Classification for Resources (UNFC). The test application to the TSF Bollrich (Germany), situated in a complex environment close to various stakeholders, demonstrates that a quick and remote assessment with publicly accessible information is possible. Since an assessment of tailings under conventional classification codes from the primary mining industry neglects relevant aspects, it is concluded that tailings should be considered as anthropogenic RMs. The developed screening approach can help to create a TSF inventory which captures project potentials and barriers comprehensively.

**Keywords:** anthropogenic raw materials; Bollrich; critical raw materials; tailings; environmental and social risks; resource management; United Nations Framework Classification for Resources (UNFC)

## 1. Introduction

Humanity faces the challenge of supplying a growing world population with electric energy while transitioning to a decarbonised electric energy generation. The construction and operation of decarbonised electric energy generation will significantly increase the demand for industrial minerals, as well as for base and high-tech metals [1–4]. However, most of the required raw materials (RMs) for the energy transition are produced outside the European Union (EU) [5]. This induces a potential supply risk which is aggravated by political conflicts, speculations on stock markets and the fact that many mineral RMs are produced in a few countries only. For instance, China is the global main producer of 24 out of 53 mineral RMs assessed by the German Raw Materials Agency (DERA) and is amongst the top 3 producers of other 11 mineral RMs [6].

For import-dependent regions as the EU [7–9], one way to decrease the supply risk is to diversify the mineral RMs sourcing. In the last two decades, there has been a growing interest in RMs recovery from waste [10–13] and the potential is vast: taking mineralised waste as an example, 624 Mt were produced in the EU in 2016, which is equivalent to 28%

of the total generated waste [14]. Part of the mineralised waste is produced by processing ores during which ore minerals are concentrated and unwanted minerals are rejected. The rejected minerals are called *tailings* and they consist of finely ground rocks and chemical additives which are often stored in tailings storage facilities (TSFs) [15]. Tailings can contain residual (non-)metalliferous minerals that can be valorised due to less efficient processing technologies of the past or because the contained minerals were not exploitable but are used as RMs in modern technologies [16–21]. Indeed, there are efforts to improve tailings-related safety by monitoring or the removal of contaminants for instance [22–24]. However, tailings still pose a risk to human health and life, the environment, and the economy; for instance by acid mine drainage (AMD) as a result of the oxidation of sulphide minerals in contact with air and water, heavy metal-laden dust emissions or structural collapse due to the often poor construction of TSFs in the past [15,25,26]. Hence, TSFs can be regarded as ecologically critical legacies with a RM potential.

Currently, the RM potential of tailings is not captured due to a general lack of data collection, and non-standardised practices in their exploration and classification [21,27–30]. In the primary mining industry, the classification of mineral RMs is a standardised practice to communicate economic viability [31]. For instance, the classification scheme by the Committee for Mineral Reserves International Reporting Standards (CRIRSCO) is globally accepted and its principles have also been applied in the exploration of tailings [32–34]. However, a systematic screening for potentially viable tailings is currently missing. Additionally, the primary mining industry is strongly driven by economic factors [35,36] so the classification standards mainly address the needs of investors [31]. Hence, the application of the CRIRSCO to tailings neglects the negative environmental impacts and social conflicts that are often associated with TSFs [37–41].

Since 1997, the United Nations Framework Classification of Resources (UNFC) has been developed to make the classification of natural mineral and energy RMs comparable. It has recently been placed in a larger context of resource management in order to support resource policies [42]; thus it contributes to coping with RMs supply risks. The advantage of the UNFC is that it considers environmental and social aspects as a project's potential key drivers beside economic ones [42]. Since 2018, a specification document is available to make the application of the UNFC to anthropogenic RMs possible [43]. The application of the UNFC to natural and anthropogenic RMs enables a consistent and comparable assessment of both RM types. This promotes a comprehensive overview of the available RMs. However, there is currently no standardised procedure for their assessment and classification [44]. A comparative case study applying CRIRSCO and UNFC principles to a metalliferous tailings deposit in Portugal demonstrates that the inclusion of environmental and social aspects can affect the classification result substantially [45].

In natural mineral RMs assessment, a mineral deposit must first be identified. A typical first step is reconnaissance exploration where an analysis at regional scale aims to identify areas of mineral occurrences that qualify for further investigation [46]. The following prospection and exploration aim to generate detailed geological knowledge [31]. In contrast, there is currently no standardised approach for project development of anthropogenic mineral RMs. The locations of TSFs are usually known but little information is available to evaluate a potentially viable project. For resource managers the question arises how to select tailings as a potentially viable RM? The exploration and inventory of TSFs to capture RMs availability requires in-depth research, stakeholder consultation, and on-site investigation, which is generally time-consuming and costly. This can be remedied with a pre-selection of potentially viable projects through screening comparable to reconnaissance exploration. This aspect has not yet been considered in the existing classification codes so there are no corresponding guidelines for a first TSF assessment and classification.

The goal of this article is to develop and test a systematic approach for a quick and efficient pre-selection of potentially viable tailings by screening in a structured UNFC-compliant manner. 5 steps are defined in order to systematically collect the necessary information. Assessment criteria are established in order to be able to carry out a first com-

pilation and interpretation of the data on metalliferous tailings. This includes geological, technological, economic, environmental, social and legal aspects. Based on the assessment result, it can be decided whether the selected TSF fulfils the criteria for further assessment including on-site exploration or whether it is to be inventoried for a future re-assessment due to a lack of information. The approach builds on remote data collection from publicly accessible internet sources, satellite images, scientific databases and thematic geoscientific maps. It is the first attempt to screen TSFs in a systematic and comprehensive manner. The TSF Bollrich (Germany) is chosen for the test application since it contains economically highly important RMs, such as $BaSO_4$, Cu, In, Pb, and Zn, and because it is located in a complex environment so that environmental and social aspects gain essential importance.

The research questions are: (1) should tailings be considered as anthropogenic RMs, (2) which information is necessary for TSF screening to reveal the driving factors and barriers for project development, and (3) can remotely assessed TSFs be classified with the current UNFC concept?

The research is structured as follows:

- considerations necessary for the UNFC's application to anthropogenic RMs
- argumentation for the consideration of tailings as anthropogenic RMs
- development of a quick and efficient UNFC-compliant approach for a systematic TSF screening
- case study on the TSF Bollrich with recommendations for further assessment
- discussion of the limitations of the developed systematic approach due to data uncertainty
- discussion of the developed approach in the context of RMs classification

## 2. Considerations for Anthropogenic Raw Materials Assessment

In this section, (1) terminology used interchangeably in the literature is defined as used in this article, (2) gaps in the current application of the UNFC to anthropogenic RMs are outlined, (3) the features of tailings in the context of natural mineral and anthropogenic RMs are analysed to outline necessary aspects that need to be considered in the assessment of tailings.

### 2.1. Key Words and Definitions

Metalliferous tailings from industrial processes are focussed and other mineralised waste (e.g., overburden, slags) is excluded. *TSF* refers to a physical structure to store tailings in and *(tailings) deposit* refers to a potential RM source. Generally, every TSF is a mineral occurrence in exploration terms and can potentially become a mineral RM deposit [47] (p. 124). *Target minerals* are intended for valorisation in contrast to the remaining *other minerals*. The categorisation depends on the intended valorisation path. *Recovery* refers to the physical tailings extraction and *tailings mining* refers to the whole process from exploration, recovery and processing to reclamation. *Screening* is defined as the first remote *study/assessment* to evaluate project potentials and barriers to select potentially viable projects for further assessment. It is comparable to *reconnaissance exploration* of natural mineral RMs.

### 2.2. Brief Introduction of the UNFC and Considerations for Its Application to Anthropogenic Raw Materials

The following description is based on Reference [42] (p. 2): the UNFC is a 'principles-based system in which products of a resource project are classified on the basis of three fundamental criteria: environmental-socio-economic viability (E), technical feasibility (F), and degree of confidence in the estimate (G), using a numerical coding system'. In a three-dimensional system (cf., Figure 1), these criteria are combined to classes with different categories (e.g., E1, E2, E3) and, where appropriate, to subcategories (e.g., E1.1). For that matter:

- the E category 'designates the degree of favourability of environmental-socio-economic conditions in establishing the viability of the project, including consideration of market prices and relevant legal, regulatory, social, environmental and contractual conditions',
- the F category 'designates the maturity of technology, studies and commitments necessary to implement the project. These projects range from early conceptual studies through to a fully developed project that is producing, and reflect standard value chain management principles',
- and the G category 'designates the degree of confidence in the estimate of the quantities of products from the project'.

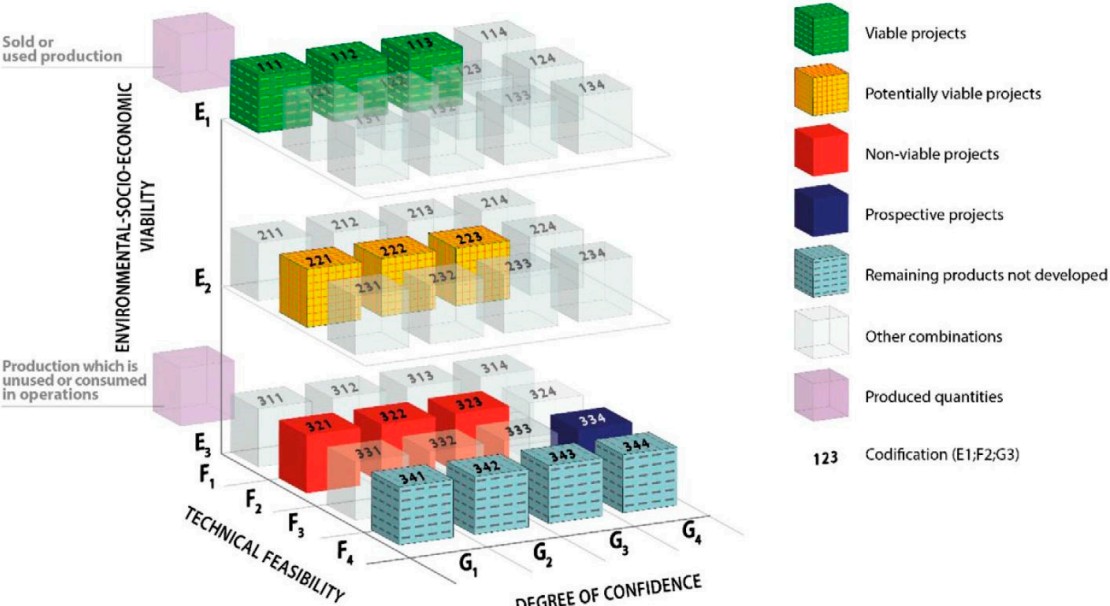

**Figure 1.** United Nations Framework Classification for Resources (UNFC) categories and examples of classes (from Update 2019 of the UNFC, by United Nations Economic Commission for Europe (UNECE) Expert Group on Resource Management (EGRM), ©2020 United Nations. Reprinted with the permission of the United Nations).

In pioneering case studies, the UNFC has been successfully applied to anthropogenic RMs, such as landfills [48], municipal waste incineration residues [49,50], electronic waste [48,51], and metalliferous tailings [45]. They deal with already identified RMs recovery projects, partially in advanced stages, showing that quality-assured data on recoverable anthropogenic RMs quantities can be evaluated with the UNFC [44]. However, the specifications document developed for its application to anthropogenic RMs merely defines relevant terminology and principles [43]. Hence, specifications are missing on how to develop a case study, which knowledge is required, and which factors and criteria should be considered for the rating of the G, F, and E categories.

An efficient resource management of anthropogenic RMs within the UNFC requires a systematic approach to identify potential projects in exploratory studies. Heuss-Aßbichler et al. [44] (pp. 7–11) make the following recommendations for the development of a sustainable resource management, which are considered in the developed approach:

- ESG issues must be addressed for the recycling of RMs,
- a broad spectrum of stakeholder perspectives must be included, and
- environmental and social impacts must be assessed and classified.

Furthermore, a first screening for a potential project should outline the requirements for further detailed investigations including the definition and characterisation phases [44] (p. 1). Moreover, it should give an overview of the potentials, barriers and relevant stakeholders.

### 2.3. Justification for the Assessment of Tailings as Anthropogenic Raw Materials

Similarities: compared to products of high purity, such as metals or complex products as mobile phones, tailings are more similar to the ores they originate from due to the relatively low degree of processing [35,52]. In the assessment, established methods from the primary mining industry can be considered for mining, the valorisation process—including mineral processing, smelting and refining—deposit modelling, and economic evaluation [32,53]. However, these aspects are not to be considered in the screening phase as they require detailed knowledge on a mineral deposit [31]. In the case of natural mineral RMs assessment, there are two types of studies which are conducted independently: first, the geological exploration of mineral RMs, which can be divided into reconnaissance, prospection, general and detailed exploration [46]. Generally, the intensity in the applied techniques and efforts increases in each phase [47]. Second, the techno-economic scoping, pre-feasibility and feasibility studies. They accompany geological exploration once reasonable prospects for an eventual economic development can be assumed [31].

Differences: in contrast to natural mineral RM deposits where the site and size are unknown in an early exploration stage, TSFs are usually close to their geogenic origin since mining operators avoid transporting material they deem valueless over long distances [47]. Hence, possible locations of TSFs can be screened for using information on active and abandoned mines. Simple methods as a visual identification of TSFs on satellite images can help to obtain basic structural information. This procedure is not applicable for finding ore deposits. Tailings consist of similar minerals as the ores they originate from so that information on local geology or on ore deposits can be used to obtain a first indication of their composition. These aspects generally enable a remote localisation and screening.

For tailings characterisation, newly generated but also historical data can be used. The targeted minerals in TSFs can vary depending on market conditions and available recovery or processing technologies. Based on the generally available data, the assessment of TSFs can be viewed as brownfield exploration [47]. Environmental and/or social aspects can also influence tailings valorisation [54]. The state of target minerals can alter in relatively short time spans due to their exposition to biological, chemical and physical processes of the Earth's surface [15]. For instance, the local climate can influence the formation of secondary minerals inside TSFs [15] (p. 172). Due to the alteration process, an inventoried TSF might need to be re-assessed in the future regarding geological conditions, its economic relevance and interested stakeholders.

In comparison to ore deposits, TSFs have an inherent negative socio-environmental impact. The severity of the individual footprint varies, depends on the condition a TSF is in and must be assessed on site [54,55]. The involved actors in a tailings mining project are partly the same as in primary mining projects: they comprise investors, mining companies, geologists, mining engineers and metallurgists for instance. However, there can be additional actors, such as modern recycling companies [56]. Furthermore, TSF owners can be the landowners if TSFs are not monitored under the Mining Law anymore [57]. Public acceptance plays a major role and depends on factors as a local population's cultural experiences [56]. Regarding the legislation, the situation can be less clear than for natural mineral RMs. For instance, tailings are considered mineral waste, thus, they fall under the Circular Economy Act (KrWG) in Germany [57]. Hence, they can only be treated in certified waste disposal plants for RMs recovery unless their legal status can be changed to a mineral RM [57]. Obtaining permission for the disposal of new residues might be a challenge too [57]. Alternatively, if environmental considerations are a project's driver, a TSF not monitored under the Mining Law would be treated under the Soil Protection Act in Germany [57]. On this basis, the legal situation must be assessed individually in more advanced studies since the uncertainties make the permitting process more costly [21]. The above aspects illustrate the complexity of the operating environments TSFs are situated in. Consequently, the screening needs to consider potential TSF-related socio-environmental impacts and a broad stakeholder group to identify project benefits and risks.

Résumé: the comparison shows that TSFs can be assessed according to the principles of natural and anthropogenic mineral RMs (cf., summary in Table 1). The natural mineral RMs approach provides the necessary geological and techno-economic information. A major difference is that in the case of anthropogenic RMs, the environmental, social and legal aspects are taken into account at an early stage, which is uncommon in natural mineral RMs exploration. Furthermore, these aspects are equally important so that all aspects must be considered concurrently to provide a comprehensive picture of the potentials of and barriers to a RM's development. These requirements can be fulfilled by assessing tailings as anthropogenic mineral RMs under consideration of the UNFC principles.

**Table 1.** General features of tailings in the context of natural mineral raw materials, consequences for the assessment of tailings, and addressed UNFC axes.

| Group & Factor | Feature | Considerations for Tailings Assessment | UNFC Axis [1] |
|---|---|---|---|
| **similarities with the assessment of natural mineral raw materials** | | | |
| **mine planning** | | | |
| mining methods | same as for ores | existing portfolio of proven methods to resort to | F |
| valorisation | same as for ores | existing portfolio of proven methods to resort to | F |
| deposit modelling | same as for ores | existing portfolio of proven methods to resort to | F |
| economic evaluation | same as for ores | existing portfolio of proven methods to resort to | E (econ.) |
| **differences from the assessment of natural mineral raw materials** | | | |
| **project identification** | | | |
| location | remnants of mining operations | mapped (non-)active mine sites can be investigated to locate TSFs | G |
| composition | similar to ore composition | first indication of tailings composition derivable from ore composition | G |
| **TSF content** | | | |
| characterisation | with historical & newly generated data | brownfield exploration: remote localisation & assessment of TSFs possible | G |
| target minerals | formerly & newly relevant raw materials | re-assessment of project viability might be necessary for inventoried TSFs | G |
| state of target minerals | can alter with time | geological re-assessment might be necessary for inventoried TSFs | G |
| **project boundaries** | | | |
| socio-environmental impact | inherent footprint of TSFs | not only geological data but also status quo impacts must be considered | E (env., soc.) |
| involved actors | broader scope of actors involved | broad stakeholder assessment necessary from screening phase on | E (soc.) |
| legislation | legal situation less clear | individual assessment necessary to clarify which laws are applicable | E (leg.) |

[1] econ.: economic aspects, env.: environmental aspects, soc.: social aspects, leg.: legal aspects.

### 3. Development of a UNFC-Compliant Approach for Systematic TSF Screening

*3.1. Concept for a Systematic TSF Screening*

While investors seek economic benefits from a project, a lack of public acceptance can jeopardise project development. Therefore, a successful project implementation must include the interest of both investors and the public. In this context, a systematic approach was developed to quickly identify project potentials and barriers in 5 steps (cf., Figure 2, elaborated in Sections 3.2–3.6). The approach implements the discussion results of Section 2.3 which stipulate to consider the principles of natural and anthropogenic mineral RMs assessment in TSF assessment. After the initial collection of basic information, the order of the steps reflects an increasing effort to obtain information. Therefore, the collection of information can be interrupted before too much time and money are invested. A reiteration of each step can be performed if it is decided that more information is necessary

or when new information on preceding steps becomes available. Table 2 shows an overview of the knowledge generated in each step, as well as general criteria for the case of a positive rating of each step.

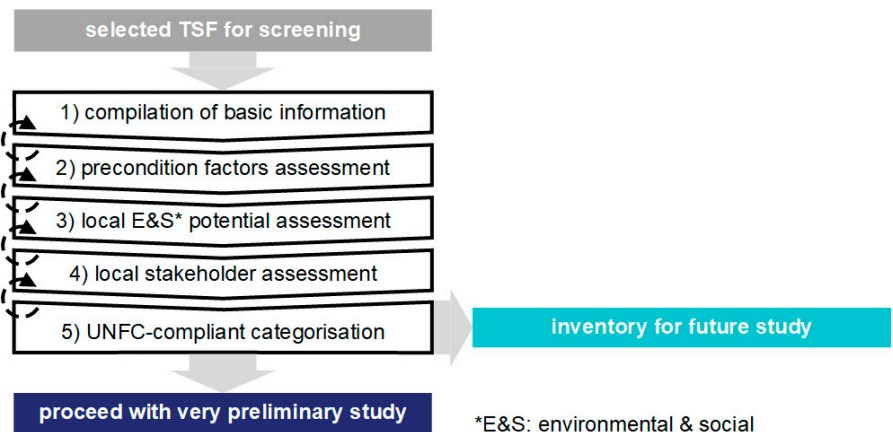

**Figure 2.** Quick and efficient UNFC-compliant approach for a systematic tailings storage facility (TSF) screening in 5 steps. The dotted lines indicate possible reiteration steps.

**Table 2.** Generated knowledge in each step and general criteria for the positive rating of each step.

| Screening Step | Generated Knowledge | General Positive Rating Criteria |
| --- | --- | --- |
| (1) basic TSF information compilation | overview is obtained, base for project definition created | all readily available basic information captured for later evaluation |
| (2) precondition factors assessment | potential project drivers identified, favourable technological & legal conditions identified | criteria of the G & F categories fulfilled, minimum one criterion of the E subcategories met |
| (3) local E&S potential assessment | possible environmental and social risks identified, potentials to reduce environmental risks and/or to create social benefits identified | minimum one conceivable positive environmental and/or social impact identified |
| (4) local stakeholder assessment | potentially affected stakeholders by TSF failure or raw materials recovery identified, potential social issues identified | all potentially affected stakeholder captured |
| (5) UNFC-compliant categorisation | generally favourable project conditions warrant on-site exploration | economic, environmental and/or social potentials/barriers identified |

The 5 steps are: (1) Basic TSF information is compiled for a general project definition. Important aspects, such as project location, environment, contained RMs, TSF condition, and potential negative impacts, are investigated. (2) The general project conditions are assessed to determine whether economic, environmental and/or social aspects could be a project's driver, and if favourable technological and investment conditions for project execution can be assumed. (3) The potential to reduce environmental and/or social risks by removing the TSF is assessed. (4) Stakeholders directly affected by the TSF or its removal are assessed and captured for a consideration in later project planning phases. (5) The

generated knowledge is reviewed and a decision regarding the further proceeding is justified with project potentials and barriers.

### 3.2. Basic TSF Information for Project Definition

Information on 3 categories is required to characterise a TSF for a first impression:

1. content
2. structure
3. location

Information on the *content* describes the geological and criticality characteristics of the tailings. It is the base to determine the tailings' economic relevance and enables an estimation of the tailings' valorisation for potentially interested stakeholders. Information on the *structure* describes a TSF's construction, technical features and current state. It is the base for a first estimation of a project's technical feasibility, expected operational risks and necessary rehabilitation measures. Information on the *location* is the basis to differentiate a TSF from its surroundings, which is necessary for describing environmental, social and infrastructural conditions.

Žibret et al. [21] propose a list of 19 factors as key basic parameters for the valorisation of mine waste, which they derived from literature and best practices reviews, as well as discussions in expert workshops. In this article, 21 basic factors are identified for the knowledge base of TSFs (cf., Table 6).

The following adaptations are made: the 21 factors are allocated to the above-described categories. A more complete impression on the TSF is obtained by adding the 12 factors raw materials, resource criticality, grade, mass, current use, local geology, topography, land use, climate, settlements, surface waters and infrastructure. At this stage, various aspects are excluded from the assessment as they require a detailed investigation (cf., Section 2.3). These are legal and permitting aspects, detailed knowledge on material- and mineral-centric valorisation parameters, and actual impacts. Potential environmental and social impacts are addressed in the environmental and social (E&S) potential assessment. The methodology of data collection and availability are presented in the case study in Section 4.

### 3.3. Precondition Factors Assessment to Identify Potential Project Drivers

Little information is available in the TSF screening phase. Therefore, the preconditions for project development are determined with 6 basic TSF information factors and 1 legal factor (investment conditions) (cf., Table 3). All factors can be allocated to the UNFC's G, F, and E categories.

**Table 3.** Precondition factors, assessed aspects, and addressed UNFC axes.

| Precondition Factor | Assessed Aspect | UNFC Axis [1] |
|---|---|---|
| (1) TSF volume | justification for mid- to long-term investment | G |
| (2) local infrastructure | cost savings due to accessible infrastructure or incurred costs due to necessary disposal of existing infrastructure | F |
| (3) TSF condition | necessity of special safety measures during mining or extensive environmental rehabilitation due to contamination | F |
| (4) resource criticality | economic importance of targeted minerals | E (econ.) |
| (5) climatic conditions | enhanced environmental risks due to TSF's location | E (env.) |
| (6) proximity to human settlements | necessity of special protective measures during mining | E (soc.) |
| (7) investment conditions | general regulatory conditions in a country | E (leg.) |

[1] econ.: economic aspects, env.: environmental aspects, soc.: social aspects, leg.: legal aspects.

G category: to attract investors, a project must be large enough to justify the investment. However, there is no empirical data available on required criteria for a tailings mining project to be viable. Hence, the factor TSF volume is chosen to address this aspect and the minimum volume is defined as 0.2 million m$^3$. It is derived from the assumption of a minimum Life of Mine (LOM) for mining metalliferous ores of 5 years (American plc (2013) cited in Reference [58]), the tonnage according to the Taylor's rule [47] (p. 320) and an average tailings density of 2 t/m$^3$ [30]).

F category: the factor local infrastructure addresses locally available technology, buildings and transportation infrastructure, and the proximity to accessible utilities infrastructure. The aim is to identify potential cost savings due to accessible infrastructure or costs due to necessary asset disposal.

The factor condition addresses project risks associated with the TSF. The aim is to anticipate costs which might be incurred due to enhanced safety measures during RMs recovery or extensive environmental rehabilitation.

E category: the E subcategories economic, environmental, social and legal aspects are addressed separately; *legal aspects* being defined as a separate E subcategory in this article.

Resource criticality of target minerals is an important aspect to assess the tailings' economic relevance. Hence, information on Critical Raw Materials (CRMs) or other RMs with very high economic importance as defined by the European Commission (EC) [59] is sought. Such RMs are often used in high-technology industries, e.g., in decarbonised electric energy generation [3]. This factor is chosen since a reliable economic estimate cannot be made without detailed geological information.

The climatic conditions give an indication on environmental risks associated with a TSF. It is an important factor as it can aggravate already existing risks: for instance, dust emissions from a TSF are more likely in arid regions, and extreme weather occurrences, such as heavy rainfalls, can erode a TSF or increase the likelihood of TSF collapse especially in combination with seismic activities. This factor is also to be considered to reduce risks in case new residues need to be disposed of locally.

The factor proximity to human settlements gives an indication if special attention must be paid during mining to protect local population, e.g., from emissions. This factor needs to be considered in the context of the climatic and TSF conditions since both can increase potential risks.

The factor investment conditions is important to indicate if simple regulations and strong protection of property rights can be expected in a country. It is assessed with a country's rank on the Ease of Doing Business ranking by the World Bank [60]. The ranking covers 12 areas of business regulation, for instance getting electricity, getting credit, and enforcing contracts [60].

### 3.4. Local Environmental and Social Potential Assessment to Identify Benefits and Risks

In general, base metal grades in tailings are low and the processing is challenging so that potential projects can be economically unviable [30]. However, environmental and social benefits can be a key driver for developing a project in anthropogenic RMs recovery [42]. The removal of a high-risk TSF represents a social and environmental advantage since it usually incurs high ecological and social costs in the long run [61].

To reveal high-risk TSFs and to assess the benefits of their removal, a local E&S risk assessment is performed (cf., Table 4). It is based on the methodology of Owen et al. [38] which was developed to assess the vulnerability of the area surrounding a TSF to its potential failure. In this article, the methodology is applied to TSF removal. The reduction of identified E&S risks is regarded as a socially responsible action; hence, it produces benefits for society.

**Table 4.** Assessed environmental and social (E&S) categories, benefits derived from TSF removal, and addressed UNFC axes.

| Category | Derived Benefits from TSF Removal | UNFC Axis [1] |
|---|---|---|
| (1) waste | reduced exposure to potential tailings flood by TSF collapse | E (env.) |
| (2) water | reduced risks to scarce water, aquatic ecosystems & drinking water | E (env.) |
| (3) landscape | reduced risk to ecosystems, aesthetically valuable lands & recreational lands | E (env.) |
| (4) biodiversity | reduced risk to nearby ecosystems | E (env.) |
| (5) land use | reduced social tensions due to land use conflicts | E (soc.) |
| (6) social vulnerability | reduced risk of harm to human health & social unrest | E (soc.) |

[1] env.: environmental aspects, soc.: social aspects.

The categories waste, water, biodiversity, land use and social vulnerability are adopted; which are described in Owen et al. [38]. The category *landscape* is added due to the importance of protected landscapes for flora and fauna, their cultural-historical significance or their values for recreation [62].

The criteria seismic hazard, aqueduct water risk, Fragile States Index and human footprint are adopted (cf., Table 8). The criterion indigenous peoples is replaced by proximity to human settlements to consider the impacts on any local population. It provides an indication of the necessity to act to protect human health since local population may potentially or may already be affected by a TSF. The criterion nearby surface waters is added to consider their exposure to a potential TSF failure. The criteria nearby nature conservation areas, water protection areas and protected landscape areas are added to consider national environmental protection regulations.

### 3.5. Local Stakeholder Assessment to Identify Potential Social Issues

The increasing importance of stakeholders in mining projects and mine site remediation is generally acknowledged [63–65]. Even more, it is increasingly recognised that social conflicts can significantly increase costs and even impede project development [66,67]. The goal of the stakeholder assessment is to identify stakeholders who must be considered in further project planning. This aspect is particularly important for investors who must be aware of social conflict potentials. 5 stakeholder categories, adapted from Azapagic [63] and Valenta et al. [65], are considered (cf., Table 5).

**Table 5.** Stakeholder categories, their selection criteria, and addressed UNFC axes.

| Category | Selection Criterion | UNFC Axis [1] |
|---|---|---|
| (1) nearby communities | potentially economically or physically affected by TSF failure or mining | E (soc.) |
| (2) TSF owner | approval required | E (soc.) |
| (3) local authorities | approval required, representing certain political interests which are relevant for tailings valorisation | E (soc.) |
| (4) NGOs [2] | representing environmental and/or social interest associated with TSF failure or tailings mining | E (soc.) |
| (5) other interested parties | any of the above | E (soc.) |

[1] soc.: social aspects. [2] NGO: non-governmental organisation.

### 3.6. UNFC-Compliant Categorisation and Final Decision

For the categorisation of the project, the knowledge on the TSF, which is generated in the previous steps, is reviewed. The results are discussed on an individual basis depending on the user's point of view. For instance, a public entity might screen a particular region for

TSFs with high environmental impacts to appraise the required environmental remediation measures. The compilation of the identified potentials and barriers together with the criteria for the removal of the barriers serves as a decision-making aid for proceeding with a very preliminary assessment. There are 2 options for a first UNFC-compliant categorisation and classification:

Proceed with very preliminary study: if the criteria outlined in Table 2 are met, the project's further assessment is recommended and the project is classified as a 'Prospective Project' in the UNFC category E3F3G4 [42] (p. 5) (cf., Figure 1). Hence, the generation of further knowledge by on-site exploration is recommended.

Inventory for future study: however, if no further assessment is recommended, the project is inventoried with the classification as 'Remaining products not developed from prospective studies' in the UNFC categorisation E3F4G4 [42] (p. 5) (cf., Figure 1).

## 4. Case Study Results

The developed approach is tested with the case study TSF Bollrich near Goslar (Germany) (cf., Figure 3). The screening is undertaken for an area downstream of the TSF within a radius of 10 km around the TSF. It is assumed that this area would be immediately threatened in case of TSF failure [38]. Moreover, it is assumed that the TSF has not yet been explored. For this reason, the various scientific studies, media reports on the TSF, and on-site exploration results [56,68–70] are excluded.

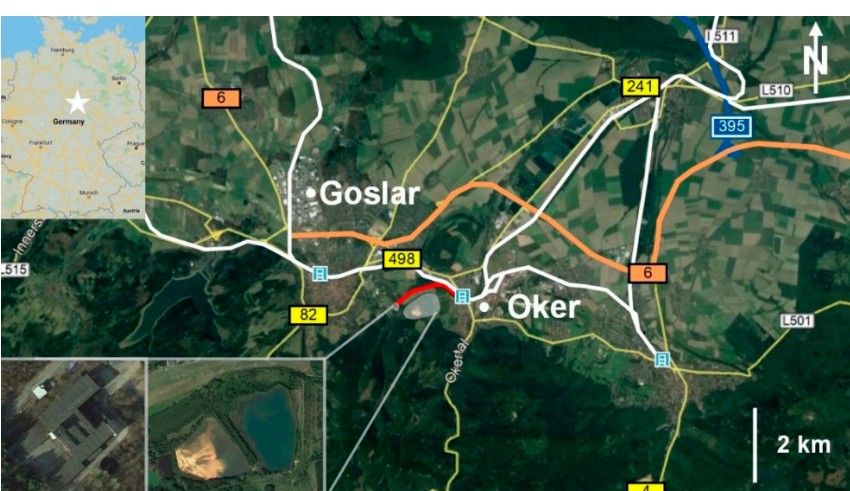

**Figure 3.** Location of the TSF Bollrich and the associated disused processing plant (light shaded areas, bottom left pictures), and public infrastructure. The white lines represent public railway tracks, the red line represents the disused railway to the processing plant Bollrich, the yellow lines represent country roads, the orange line represents the 4-lane section of the federal highway B6, and the blue line represents the motorway A395 (adapted after Google Earth [71]).

### 4.1. Basic TSF Information

The results of step 1 are summarised in Table 6, showing that information for 20 out of 21 factors could be retrieved. The data quality is rated according to the following criteria: obvious or well-documented data is rated high quality, and remotely obtained data requiring exact data, speculative or indirect data is rated low quality. The primary sources of information are a combination of observations on Google Earth [71] and a Google search which evolved from the observations.

G category: regarding the geological evidence, most data is based on indirect evidence so the data quality is accordingly low: the presence of $BaSO_4$, Ag, Au, Cu, In, Pb, and Zn is only assumed based on the composition of the mined ores in References [72,73] and the description of ore processing by Eichhorn [74]. Several changes in the ore processing during the TSF's operation are described [74] so that variations in mineral quantity, quality

and distribution can be expected. The TSF volume is a rough estimate based on a sketch with AutoCAD, the contents of the 3 ponds cannot be differentiated, and mineral quantities and qualities are missing. There is no information regarding the neutralised mine waters in the middle pond, and the quantity and composition of the discharged residues.

F category: public infrastructure, such as roads, motorways, highways, and railway tracks, are in near vicinity of the TSF. It is observable on Google Earth that the TSF is accessible via dirt roads and a disused railway track connects the processing plant Bollrich to the public railway network in Oker (cf., Figure 3). It can be observed that the railway track is partly overgrown by vegetation (cf., coordinates: 51°54′15.68″ N, 10°27′17.66″ E). This is confirmed by photos retrieved from an internet forum (http://www.goslarer-geschichten.de/showthread.php?2000-Regelspurige-Erzbahn-Bollrich-nach-Oker), also showing that the wooden railway sleepers are partly rotten. It is also observed that the buildings of the processing plant still exist.

Overall, most factors could be investigated with high quality data and only the factor *grade* lacks information. Based on the basic TSF information, the assessment is continued.

### 4.2. Precondition Factors Assessment

The results of step 2 are summarised in Table 7, showing that 6 out of 7 criteria are rated positive. The sources of information are scientific publications, public databases and observations on Google Earth [71].

It can be assumed that despite the simple estimation, the minimum TSF volume is exceeded 20-fold. Buildings, transportation and utilities infrastructure is present and it is assumed that all are accessible and might be reused. Erosion of the TSF or other problematic conditions, such as AMD, are not observable so that risks to a mining operation are assumed to be low and no major environmental rehabilitation measures can be anticipated. The presence of the CRMs $BaSO_4$ and In, and the economically highly important elements Cu, Pb, and Zn make the TSF economically interesting. A low climatic risk can be assumed so that related risks to a mining operation or the locally disposed of new residues are unlikely. The TSF's proximity of approximately 400 m to the nearest human settlement is rated critical. As for the investment conditions, Germany has a very high rating on the Ease of Doing Business ranking, so that favourable regulatory conditions for project execution are assumed. In summary, the project preconditions are rated favourable so that an investor's interest in the TSF can be justified.

### 4.3. Local Environmental and Social Potential Assessment

The results of step 3 are summarised in Table 8. The indicator thresholds are chosen conservatively to capture high risks only. The sources of information are public scientific and non-scientific databases, as well as published reports. An overview of the TSF's near environment in the context of environmentally sensitive areas is given in Figure 4.

A visual assessment of the tailings flow direction in case of a dam breach was performed with a topographic map (cf., Figure A1). It shows that the flanks of the valley in which the TSF has been built form a funnel which would direct the tailings towards the public railway tracks and the nearby industrial area in Oker. When conservatively assuming a flow rate of 5 km/h [38], the tailings would reach the nearest observable buildings on Google Earth and the public railway tracks in approximately 1.2 min (100 m distance) and 5.3 min (440 m distance), respectively. It is doubtable that the area could be evacuated in such a short time span so that harm to human health would be likely. A tailings spill could also affect the protected landscape area downstream of the TSF (cf., Figure 4).

The proximity to the river Gelmke, which flows immediately downstream of the TSF, is critical. Due to the river's small size, a tailings spill would completely fill up the river, destroy the aquatic ecosystem and deprive the river of its drain.

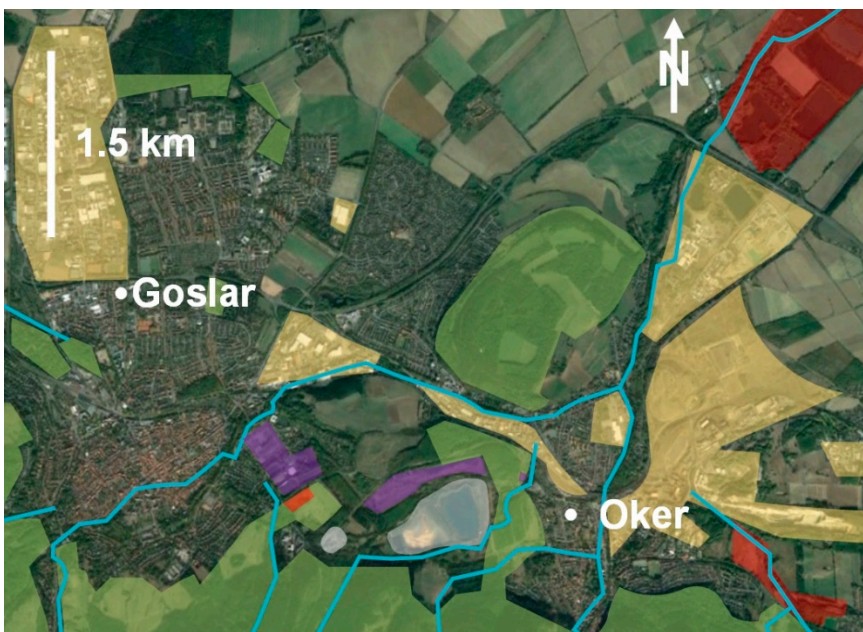

**Figure 4.** Simplified schematic illustration of the environment around the TSF Bollrich: the light grey shaded areas mark the TSF Bollrich (right area) and the associated disused processing plant (left area), the green shaded areas mark protected landscape areas, the red shaded areas mark nature conservation areas, the yellow shaded areas mark industrial and commercial areas, and the purple shaded areas mark sports areas close to the TSF. The blue lines represent rivers (adapted after District of Goslar | Environmental Service [75] and Google Earth [71]).

The area around the TSF is strongly affected by human activity with a Human Footprint Index of 60–80% so that there could be competing land use interests. The city administration of Goslar has the goal to develop the area around the TSF as an extensive natural and cultural landscape for calm recreation [76]. Hence, the removal of the TSF could contribute to fulfilling this goal, for instance by restoring a more natural environment.

In contrast, the seismic risk is relatively low and no signs of dam erosion are observable on Google Earth so that the risk of TSF failure is rated low. The water risk is low so that local water supply is assumed not to be endangered in case of TSF failure and a competition between different water users is unlikely. The spotted water protection and nature conservation areas downstream of the TSF is assumed not to be immediately threatened by a tailings spill due to the region's topography and the distance (cf., Figure 4). The social indicators give rise to the assumption that local communities would be able to cope with TSF failure.

Overall, the assessment of the environmental and social risk categories reveals that the TSF's environment is vulnerable to a possible TSF failure. Its removal would, therefore, generate benefits despite the low risk of failure. The proximity to human settlements and the complex surrounding environment are regarded as a necessity to act.

*4.4. Local Stakeholder Assessment*

The results of step 4 are depicted in Table 9, listing 17 stakeholders or stakeholder groups that could be identified. The primary sources of information are a combination of observations on Google Earth [71], a Google search with related and various other search terms, and a published report of an integrated development concept for the city Goslar by Ackers and Pechmann [76].

The largest stakeholder group consists of the citizens of Goslar and Oker with approximately 50,000 inhabitants in total [76]. Regarding local authorities, the State Office for Mining in Clausthal-Zellerfeld and various departments of the city administration of Goslar have to be considered for regulatory aspects. Three local environmental non-governmental

organisations (NGOs) could be captured, counting more than 1800 members according to their websites. Their early consultation is crucial to obtain public acceptance for project implementation in an environment highly impacted by industrial activities. The German Railway and the company Oker-Chemie are directly threatened by a possible TSF failure and could advocate its removal. Farmers, foresters, and the air sports community surrounding the TSF need to be protected from negative impacts during mining. The development association of the Rammelsberg mine preserves the cultural heritage and it needs to be assured that a project would not contradict their interests. The Clausthal University of Technology and the Recycling Cluster Economically Strategic Metals (REWIMET e. V.) should be considered for their experience with recycling technologies and mine waste valorisation.

No information could be retrieved on the TSF's owner. 3 stakeholders could be captured but not specified: the responsible entity for the discharge of mine water from the Rammelsberg mine into the TSF, the owner of a tennis court and a company located approximately 230 m downstream of the TSF.

In summary, the assessment shows that the TSF is situated in a complex environment due to its proximity to agricultural, forest, industrial and commercial, nature and water protection, recreation, and residential areas (cf., Figure 4). In this context, a comprehensive stakeholder management is recommended if the project is to be continued.

### 4.5. UNFC-Compliant Categorisation

Overall, a further assessment within the scope of a very preliminary study is recommended due to the following aspects which are favourable for the development of a tailings mining project: assumed presence of CRMs and economically highly important metals, the identified potentials of environmental risk reduction and benefits of environmental rehabilitation, the potential to reduce land use-related social tension, favourable regulatory and infrastructure conditions, and a sufficiently large TSF volume. According to the UNFC [42], the project is classified as a 'Prospective Project' in the E3F3G4 categorisation (cf., Figure 1).

### 4.6. Path Forward for the Case Study Bollrich

In a very preliminary study, the following aspects should be addressed to remove the barriers for a higher classification as a 'Potentially Viable Project' (E2F2G3): the largest barrier is the lack of geological knowledge on the deposit so that, for instance, the quantities of products cannot be estimated. Hence, the next milestone is on-site exploration to determine material characteristics, such as the chemical and mineralogical composition of the tailings, their quantities and qualities, and their physico-chemical properties, as well as their distribution inside the TSF. Additionally, the TSF's geomechanical stability needs to be studied. Furthermore, the identification of the TSF's owner and a first assessment of the legal conditions for a project are important aspects to be clarified.

**Table 6.** Basic information on the TSF Bollrich. The green shaded and red shaded shaded areas indicate data of high and low quality, respectively.

| Category & Factor | Data | Source & Data Quality |
|---|---|---|
| **(1) content** | | |
| (i) raw materials | sulphates: $BaSO_4$; sulphides: Cu, Pb, Fe, Zn; others: Ag, Au, In | inferred from References [72,73] |
| (ii) resource criticality | $BaSO_4$, & In are Critical Raw Materials in the EU; Cu, Pb, & Zn of very high economic importance in the EU | [59] |
| (iii) grade | - | - |
| **(2) structure** | | |
| (iv) history | start/end of operation in 1938/1988, froth flotation plant Bollrich closed in 1987, course of Gelmke was modified several times | [74] |
| (v) reasons for closure | closure of mine Rammelsberg in 1988 for economic reasons | [74] |

**Table 6.** *Cont.*

| Category & Factor | Data | Source & Data Quality |
|---|---|---|
| (vi) design | valley impoundment, 1 small pond & 2 large ponds, 1 main dam & 2 intermediate dams, estimated dam height 35 m | observed on Google Earth [71], cf., Figure 3 |
| (xii) surface area | estimated 315,000 m$^2$ | Ruler tool [74] |
| (xiii) volume | estimated 4.7 & 4 million m$^3$ (including & excluding main dam, respectively) | Ruler tool [74], AutoCAD (Autodesk Inc.) |
| (iv) mass | estimated 9.4 & 8 million t (including & excluding main dam, respectively) | assumed tailings density 2 t/m$^2$ [30] |
| (x) homogeneity | several changes of ore processing reported, heterogeneity of minerals inside TSF can be assumed | [74] |
| (xi) condition | partially dry but mostly covered with water, no observable signs of AMD, erosion or controlled reclamation | observed on Google Earth [71], cf. |
| (xii) current use | since 1988 neutralised mine waters from the closed mine Rammelsberg are discharged into the lower pond | observed on Google Earth [71,74] |
| **(3) location** | | |
| (xiii) position | Goslar district (51°54′8.97″ N, 10°27′47.31″ E, Lower Saxony, Germany), 270 m above mean sea level | observed on Google Earth [71] |
| (xiv) local geology | folded & faulted Palaeozoic rocks of the Harz Mountains are uplifted & thrust over younger Mesozoic rocks of the Harz foreland along the Northern Harz Boundary fault leading to steeply tilting & partly inverted Mesozoic strata, Mesozoic rocks are largely composed of Triassic to Cretaceous sedimentary rocks of varying composition (i.e., mostly impure limestones, clastic sandstones (greywackes) & shales), younger Quaternary sediments are rare & locally limited | [77] |
| (xv) topography | at the foot of Harz Mountain range, max. 1141 m altitude with deep valleys | [78] |
| (xvi) land use | in near vicinity: agricultural, forest, industrial & commercial, & recreation & residential areas | observed on Google Earth [71] |
| (xvii) climate | moderately warm, temperature −0.7 to 16.3 °C (average 7.9 °C), average rain precipitation 768 mm/a | [79] |
| (xviii) settlements | nearest ~400 m E air-line distance downstream of main dam | observed on Google Earth [71], cf., Figure 3 |
| (xix) surface waters | 4 small rivers observed downstream of TSF within 1.5 km radius (Abzucht, Ammentalbach, Gelmke, Oker) | observed on Google Earth [71], cf., Figure 4 |
| (xx) site accessibility | dirt roads, federal highway B6 ~1.6 km N air-line distance from TSF, public railway ~500 m E air-line distance from TSF, disused railway tracks from processing plant Bollrich to public railway network (estimated abandonment in 1988) | observed on Google Earth [71,74] cf., Figure 3 |
| (xxi) infrastructure | disused processing plant Bollrich ~500 m W air-line distance from TSF, access to public electricity & water grid assumed | observed on Google Earth [71], cf., Figure 3 |

**Table 7.** Precondition factors, and the corresponding criteria and indicators for a TSF screening. ✓ indicates a fulfilled and ✗ a non-fulfilled criterion, respectively.

| Factor | Criterion | Indicator | Result | Source | Rating | UNFC Axis [1] |
|---|---|---|---|---|---|---|
| (1) TSF volume | TSF volume (V) high enough for a LOM [2] of ≥ 5 years | $V \geq 0.2$ million m$^3$ | 4 million m$^3$ (excluding main dam) | estimated with Ruler tool in Google Earth [71] & AutoCAD (Autodesk Inc.) | ✓ | G |
| (2) infrastructure | buildings, transportation & utilities infrastructure present | observable | buildings, railway tracks, roads, highways, motorways & utilities infrastructure observable | assumption based on observation with Google Earth [71] | ✓ | F |

**Table 7.** *Cont.*

| Factor | Criterion | Indicator | Result | Source | Rating | UNFC Axis [1] |
|---|---|---|---|---|---|---|
| (3) TSF condition | erosion of TSF and/or emissions (e.g., AMD [3]) | not observable | no signs of erosion and/or emissions observable | observation with Google Earth [71] | ✓ | F |
| (4) resource criticality | number ($n$) of elements or minerals that are CRMs [4] in EU or that are of very high economic importance | $n \geq 1$ | $n = 4$ ($BaSO_4$, Cu, Pb & Zn expected to be present) | inferred from [73] | ✓ | E (econ.) |
| (5) climatic conditions | favourable climatic conditions with low probability of extreme climate or weather occurrences | moderate climate | moderately warm, average 7.9 °C, average rain precipitation 768 mm/a | [79] | ✓ | E (env.) |
| (6) human settlements | distance ($d$) to settlements | $d \leq 10$ km | $d \approx 400$ m E air-line | [71] | ✗ | E (soc.) |
| (7) investment conditions | good conditions as per Ease of Doing Business ranking | country rank ≤ 75 | rank 22 (Germany) | [60] | ✓ | E (leg.) |

[1] econ.: economic aspects, env.: environmental aspects, soc.: social aspects, leg.: legal aspects. [2] LOM: Life of Mine. [3] AMD: Acid Mine Drainage. [4] CRM: Critical Raw Mater.

**Table 8.** Results of the local E&S potentials assessment for the TSF Bollrich (modified after Owen et al. [38]). ✓indicates a fulfilled and ✗ a non-fulfilled criterion, respectively.

| Domain [1] | Category | Criterion | Indicator | Result | Source | Rating |
|---|---|---|---|---|---|---|
| env. | waste | seismic hazard | peak ground acceleration > 3.2 m/s² | 0.4 m/s² | [80] | ✗ |
| | water | aqueduct water risk | overall water risk > 3 (high) | 1–2 (low-medium) | [81] | ✗ |
| | | nearby surface waters | downstream distance to TSF < 10 km | in near vicinity, cf., Figure 4 | [71] | ✓ |
| | | nearby water protection areas | downstream distance to TSF < 10 km | ~7.3 km N-E of the TSF near Vienenburg | [75] | ✓ |
| | landscape | protected landscape areas | downstream distance to TSF < 10 km | nearest immediately at the foot of the dam, cf., Figure 4 | [75] | ✓ |
| | biodiversity | nature conservation areas | downstream distance to TSF < 10 km | ~3.5 km N-E of TSF, cf., Figure 4 | [75] | ✓ |
| soc. | social vulnerability | proximity to human settlements | downstream distance to TSF < 10 km | nearest settlement Oker ~400 m E of main dam, potential flow path in direction of settlement, cf., Figure A1 | [71,82] | ✓ |
| | | Fragile States Index | country score ≥ 4 for social indicators | average score 2 (Germany) | [83] | ✗ |
| | land use | human footprint | Human Footprint Index > 40% | 60–80% (area around the TSF) | [84] | ✓ |

[1] env.: environmental potentials, soc.: social potentials.

**Table 9.** Potential stakeholders of a tailings mining project at the TSF Bollrich (stakeholder categories derived from Azapagic [63] and Valenta et al. [65]).

| Stakeholder Category | Result | Source | Remark |
|---|---|---|---|
| nearby communities | (1) citizens of Goslar & its borough Oker | observation on Google Earth [71,76] | total population of ~50,000 inhabitants |
| TSF owner | (2) - | - | could not be clarified with internet search |
| local authorities | (3) Goslar administrative bodies | www.landkreis-goslar.de www.landkreis-goslar.de/eh- | Various departments, such as for Regional Economic Development or the Environment, the Circular Economy Department, are responsible for the disused landfill Paradiesgrund in near vicinity of the TSF |
| | (4) State Office for Mining, Energy & Geology Office Clausthal-Zellerfeld | www.lbeg.niedersachsen.de | ~15 km S-W from TSF, included due to relevance for approval |
| NGOs | (5) German Federation for the Environment & Nature Conservation in the western Harz region (BUND) | www.bund-westharz.de | ~600 members |
| | (6) Nature & Biodiversity Conservation Union (NABU) | www.nabu-goslar.de | ~1000 members |
| | (7) Nature & Environmental Aid Goslar (NU) | www.nu-goslar.de | ~200 members |
| other interested parties | (8) German Railway (DB) | observation on Google Earth [71] | connection to railway network would potentially have to be reactivated, a potential TSF failure might affect the railway |
| | (9) farmers | observation on Google Earth [71] | proximity to farmlands around the TSF |
| | (10) foresters | observation on Google Earth [71] | proximity to forests around the TSF |
| | (11) Development Association World Cultural Heritage Ore Mine Rammelsberg Goslar/Harz | https://foerderverein-rammelsberg.de | the association is responsible for the preservation of the World Heritage |
| | (12) Oker-Chemie GmbH | observation on Google Earth [71] | a potential TSF failure might affect the industrial site |
| | (13) Air Sports Community Goslar | www.segelfliegen-goslar.de | glider airfield in near vicinity of TSF |
| | (14) REWIMET e. V.—Recycling Cluster | www.rewimet.de | network of companies, scientific institutions & local authorities, promotes recycling from research up to the industrial scale |
| | (15) Clausthal University of Technology (TUC) | www.ifa.tu-clausthal.de | ~14 km S-W from TSF, included due to regional knowledge & research experience on mineral wastes of >25 years |

**Table 9.** *Cont.*

| Stakeholder Category | Result | Source | Remark |
|---|---|---|---|
| | non-specifiable: | | |
| | (16) responsible entity for mine water discharge into the TSF | observation on Google Earth [71] | could not be specified with internet search |
| | (17) owner of tennis courts downstream of the TSF | observation on Google Earth [71] | could not be specified with internet search |
| | (18) company downstream of the TSF | observation on Google Earth [71] | could not be specified with internet search |

As for the technical feasibility, different valorisation scenarios should be investigated. Hence, the tailings' processability needs to be assessed together with a conceptual mine plan under consideration of various valorisation options. This includes an investigation of the decommissioned Bollrich processing plant, whether there is reusable machinery, and the condition of the road and railway access.

This article shows that a large, diverse and socially active stakeholder group is involved. Therefore, early proactive stakeholder engagement is recommended. Measures should be taken to avoid negative environmental impacts on local population during active mining to avoid social conflicts. A public discussion of the benefits and risks of the status quo of the TSF can help to promote public acceptance. A strong argument for removing the TSF is the risk of greater harm in the event of TSF collapse. In this case, an expansion of the 10 km screening radius could help to better estimate potential harm and determine whether additional stakeholders would be affected. A detailed survey of actual emissions from the TSF could provide additional arguments for its removal.

Economic and social aspects of the city administration's development goals can also contribute to the evaluation: strengthening the regional industrial and commercial role, creating high-value jobs, fostering cultural heritage and traditions, harnessing the cultural potential of the industrial history, and developing tourism [76]. The likelihood of obtaining political acceptance increases if possible RMs recovery scenarios do not contradict these goals. Additionally, the likelihood of obtaining political and public acceptance can be increased if part of the revenues from a project would be used for partial environmental rehabilitation of contaminated land in Oker [76].

## 5. Discussion

### 5.1. Limitations of the Developed Screening Approach

Data sources and quality: the developed approach is intended to enable a quick and cost-efficient TSF screening. This goal can be achieved as shown by the case study application. A low degree of data quality can be tolerated in the screening phase similar as in reconnaissance exploration [47]. However, one needs to be aware of the potential sources of error: the information's quality from publicly accessible sources can vary from speculative (e.g., private websites or internet forums) to scientifically proven (e.g., peer-reviewed articles), and cross-checking is not always possible. In internet forums and websites, participants generally do not reflect representative interest groups, and the opinions shared may be biased.

The methods used can generally be expected to provide low quality data on a TSF's content and structure. Tailings production records or exploration data are unlikely to be publicly accessible especially for older TSFs. The visual assessment of satellite images can only provide a rough estimate of a TSF's volume and water body. Statements about the dam material or other materials inside a TSF cannot be made.

Certain aspects can only be hardly or not evaluated at all with Google Earth, such as the condition of present infrastructure. A visual assessment and internet search are unsuitable for carrying out a comprehensive stakeholder analysis. In the case study, for

instance, there are unspecifiable stakeholders, such as the TSF's owner. In the early stage of project development, no tendencies towards public acceptance can be anticipated.

The E&S potential assessment is generally expected to generate information of high quality since it relies on established scientific and public databases. However, information on certain factors might not always be available in the required quality, especially in remote areas. In addition, actual negative environmental impacts, such as emissions to air, need to be assessed on site.

Categorisation of screening results: the case study shows that the developed approach can be used to compile sufficient information for a TSF screening. The evaluation of the generated knowledge allowed to identify project potentials and barriers that need to be considered for further project development. According to the UNFC [42] (p. 5), only 2 categorisations are possible in the screening phase: a 'Prospective Projects' (E3F3G4) or 'Remaining products not developed from prospective projects' (E3F4G4).

A disadvantage of this limitation is that important aspects of a project's status cannot be communicated directly. It is, therefore, recommended to consider the following aspects for sustainable resource management of anthropogenic RMs: the sources of information, e.g., historical, indirect, or speculative, cannot be differentiated in the G subcategories. A differentiation could provide a quick overview of the information's quality. The F subcategories are not applicable since they focus on the degree of development of recovery technologies and neglect factors, such as already existing infrastructure. This is overcome in this article by assuming that present or absent observable infrastructure can be distinguished with the F3 and F4 categories. An according description should be added to the guideline for anthropogenic RMs. The E categorisation does not allow for a differentiated communication of a project's potentials and barriers in an appropriate level of detail since several dimensions are aggregated in the E category. There is the need to differentiate the E category by introducing the 4 separate subcategories economic, environmental, social, and legal aspects.

### 5.2. The Developed Screening Approach in a Global Raw Materials Classification Context

The conventional classification of tailings under the CRIRSCO has several shortcomings: first, early exploration focusses on geological aspects, such as mineral quality and quantity, in order to identify potentially economic mineral RMs [31]. Exploration Targets, Exploration Results, and non-economic mineral RM deposits are excluded from the classification [31]. However, metal grades in tailings are generally low [30] so that, from the CRIRSCO's perspective, the exploration of TSFs can be expected to be a priori unattractive due to the high costs. In contrast, the developed approach shows that it is possible to perform a quick TSF screening and a UNFC-compliant categorisation with little effort in order to determine a TSF's potentials. This quick and efficient approach can help make TSF exploration more attractive.

Second, the CRIRSCO generally focusses on providing information for investors [31]. Hence, the definition of a RM's potential under the CRIRSCO is limited to material and monetary aspects. However, when assessing mineral RMs, aspects other but purely economic ones are increasingly becoming important [65], and environmental, social, and legal aspects must be taken into account explicitly in the case of anthropogenic RMs. As shown by the case study, the latter aspects can even be decisive for the screening result. It also shows that these aspects can be assessed parallel to geological and economic ones unlike it is standard practice in natural mineral RMs assessment.

Third, under the CRIRSCO, sustainability aspects are discussed in Public Reports but they are not relevant at exploration stage and they are not part of the classification [31]. However, the physical risks of TSFs are often borne by local populations and the environment while mining companies mostly face financial risks only [38]. Communities living in near vicinity to TSFs are often unable to properly judge the risks associated with TSFs since these are rarely disclosed [38]. This is particularly important since the communities can usually not move away to avoid these risks [38]. Currently, resource management recog-

nises that E&S risks can form a barrier to mining projects [65]. Therefore, the assessment of relevant stakeholders, including public institutions and local communities, is important even in an early exploration phase. Furthermore, identified TSF-related environmental and social impacts at local level must be incorporated into a resource strategy at an early stage. Additionally, the awareness of cultural factors must be included especially in the development of anthropogenic RMs since they can enable or prevent their valorisation [56]. Overall, these aspects would ensure a high level of transparency for all stakeholders.

Fourth, the environmental context a RM deposit is situated in is not considered when reporting Exploration Targets or Exploration Results [31]. This includes aspects such as already present infrastructure and risks. However, it could be shown in this article that information on these aspects is remotely retrievable with low effort and that it can provide important information on a project's potentials and barriers.

Fifth, the assessment and reporting criteria defined in the CRIRSCO apply to Exploration Results and more advanced studies only [31]. Several authors state that there is a lack of knowledge on TSFs and their properties [29,44,85] so that the overall risks and economic potentials remain generally unknown. In the EU, for instance, current mine waste inventories are not comprehensive: the 'Minerals4EU' Knowledge Data Platform (http://www.minerals4eu.eu) provides too little information, and national mine waste registries of the EU's countries are incomplete and focus on environmental aspects in most cases [21]. To remedy this shortcoming, a comprehensive analysis of the status is needed, including economic, environmental, social, legal, technological, and geological aspects altogether. A status analysis would enable one to filter for specific aspects, such as RMs recovery potentials. However, the CRIRSCO is unsuitable for a screening that fulfils these requirements. In contrast, the developed 5-step approach can enable a status analysis in a remote, quick, and, therefore, cost-efficient manner.

## 6. Conclusions and Recommendations

To recapitulate, anthropogenic RMs are becoming an increasingly important source of RMs and they are available in vast amounts: for instance, the world's largest waste stream, mineralised waste, is produced in an estimated range of 20–25 Gt/a [15]. The current knowledge gaps concerning their RM potential and the lack of comparability to other RMs impede valorisation. In the context of an expected increase of global demand for metallic and non-metallic RMs of 96% and 168%, respectively, between 2015 and 2050 [86], actions must be taken to include anthropogenic RMs in strategic resource management. One solution is a comprehensive investigation and sharing of the information with decision-makers and the public. Therefore, the potentials of and barriers to their development need to be mapped, comparable to current practice in natural mineral RMs assessment. This must be done in a quick and cost-efficient manner. This study uses tailings as an example to demonstrate how a UNFC-compliant approach can be used for a systematic TSF screening, similar to the concept of reconnaissance exploration for natural mineral RMs. The case study TSF Bollrich (Germany) is chosen to show how a TSF can be systematically screened in practice with the developed approach. Hence, the innovative approach contributes to a re-interpretation of the material value of tailings in terms of resource efficiency within a circular economy.

The research questions are answered: (1) Tailings are a suitable example to demonstrate the difference between natural and anthropogenic RMs. The CRIRSCO classification standards from the primary mining industry are designed for natural mineral RMs and they focus on material and monetary aspects. However, the case study shows that sustainability aspects together with legal aspects and the interest of stakeholders are of vital importance for tailings assessment. This corresponds to the general requirements for the classification of anthropogenic RMs. (2) For a comprehensive assessment of tailings, all relevant aspects must be considered including information on a TSF's contents, physical structure and surroundings. The concerns of local population on potential negative environmental and social impacts can be a major barrier to project development. Therefore, the stakeholders

who are potentially affected by a TSF and its removal should be investigated even in a screening phase. (3) Applying a UNFC-compliant approach, which considers environmental and/or social aspects in addition to economic viability, can increase the chances for RMs recovery from TSFs. A categorisation of the retrieved information is possible and a classification of 'Prospective Projects' (categorisation E3F3G4) and 'Remaining products not developed from prospective studies' (categorisation E3F4G4) can be performed with the current UNFC concept.

The case study results are summarised: the case study application shows that a quick and remote identification of a potentially viable tailings mining project is possible. It is based on geological, techno-economic, environmental, social and legal aspects. Hence, a decision for further assessment can be made before costly on-site exploration is carried out. A particularity of the case study TSF is its embedding in a complex environment in near vicinity to local population. It highlights the importance of considering TSF-related environmental and social impacts on a local scale. Potentials for project development are that a potential source of economically highly important RMs is identified, the city administration's development goals can be supported, environmental rehabilitation can be promoted, and social risks can be reduced. Barriers are the lack of a conceptual mine plan including techno-economic feasibility, high uncertainties regarding data on mineral quantity and quality, and the lack of information on actual environmental and social impacts.

The following recommendations are made: for the case study TSF Bollrich, increase the degree of confidence in knowledge on geology and technical feasibility by on-site exploration. Identify the TSF's owner and determine legal conditions for mining. Investigate stakeholder opinions to anticipate conditions for public acceptance. Develop valorisation scenarios which consider the development goals for Goslar, environmental rehabilitation, and a wide variety of stakeholder interests. Systematic screening approach: identify the requirements for a project status as 'Potentially Viable'. Investigate if mineral- and structure-related information on TSFs can be obtained with spaceborne hyperspectral and radar measurements, respectively. Develop a standard at European level, including reporting guidelines, in order to comprehensively map the RM potential of tailings. Test if governance-related risks can be included in the screening. Develop more case studies to identify essential and universally applicable valorisation factors and assessment criteria including sustainability. Test the developed approach with other mine wastes. Anthropogenic resource management: implement a screening for potentially viable recovery projects. Break down the UNFC's E category into economic, environmental, social and legal aspects to visualise specific project potentials and barriers. Develop guidelines for rating data quality and uncertainty ranges in project development stages. Include stakeholder assessments in case studies to capture potential sources of social conflict. To enable an EU-wide comparison of the RM potential of tailings and to reveal barriers for project development, structure information in EU national mine waste registries in a UNFC-compliant manner.

**Author Contributions:** Conceptualisation, R.S.; methodology, R.S.; validation, R.S., S.H.-A.; resources, R.S.; data curation, R.S.; writing—original draft preparation, R.S.; writing—review and editing, R.S., S.H.-A.; visualisation, R.S.; project administration, R.S.; funding acquisition, R.S., S.H.-A. Both authors have read and agreed to the published version of the manuscript.

**Funding:** This research was funded by the German Ministry of Research and Education (BMBF) as part of the research project ADRIANA under the Client II programme, grant agreement number 033R213A-D.

**Institutional Review Board Statement:** Not applicable.

**Informed Consent Statement:** Not applicable.

**Data Availability Statement:** This research used publicly available data available in the referenced sources.

**Acknowledgments:** The authors are thankful to Bernd G. Lottermoser for his comments and to Johannes J. Emontsbotz for assisting with the CAD modelling. In addition, the authors would like to express their deep gratitude to 4 anonymous reviewers who helped to improve the manuscript.

**Conflicts of Interest:** The authors declare no conflict of interest. The funders had no role in the design of the study; in the collection, analyses, or interpretation of data; in the writing of the manuscript, or in the decision to publish the results.

## Abbreviations

| Abbreviation/Unit | Description |
| --- | --- |
| Ag | lat. *argentum* (silver) |
| Au | lat. *aurum* (gold) |
| $BaSO_4$ | barium sulphate (barite) |
| Cu | lat. *cuprum* (copper) |
| Fe | lat. *ferrum* (iron) |
| In | indium |
| Pb | lat. *plumbum* (lead) |
| Zn | zinc |
| AMD | acid mine drainage |
| CRIRSCO | Committee for Mineral Reserves International Reporting Standards |
| CRM | Critical Raw Material |
| E | East |
| E&S | environmental and social |
| EC | European Commission |
| EU | European Union |
| LOM | Life of Mine |
| N | North |
| N-E | Northeast |
| NGO | non-governmental organisation |
| REWIMET e. V. | Recycling Cluster Economically Strategic Metals |
| RM | raw material |
| S-W | Southwest |
| TSF | tailings storage facility |
| UNFC | United Nations Framework Classification for Resources |
| UNFC E category | represents environmental-socio-economic viability |
| UNFC F category | represents technical feasibility |
| UNFC G category | represents degree of confidence in the geological estimate |
| W | West |
| °C | degree Celsius (unit of temperature on the Celsius scale) |
| Gt/a | gigatons per year (unit of mass flow, equivalent to $10^{12}$ kg per year) |
| km | kilometre (unit of length, equivalent to 1,000 metres) |
| m | metre (SI unit of length) |
| $m/s^2$ | metre per square second (unit of acceleration) |
| $m^3$ | cubic metre (SI-derived unit of volume) |
| mm/a | millimetres per year (annual rain precipitation) |
| t | metric tonne (unit of weight, equivalent to 1,000 kg) |

**Appendix A**

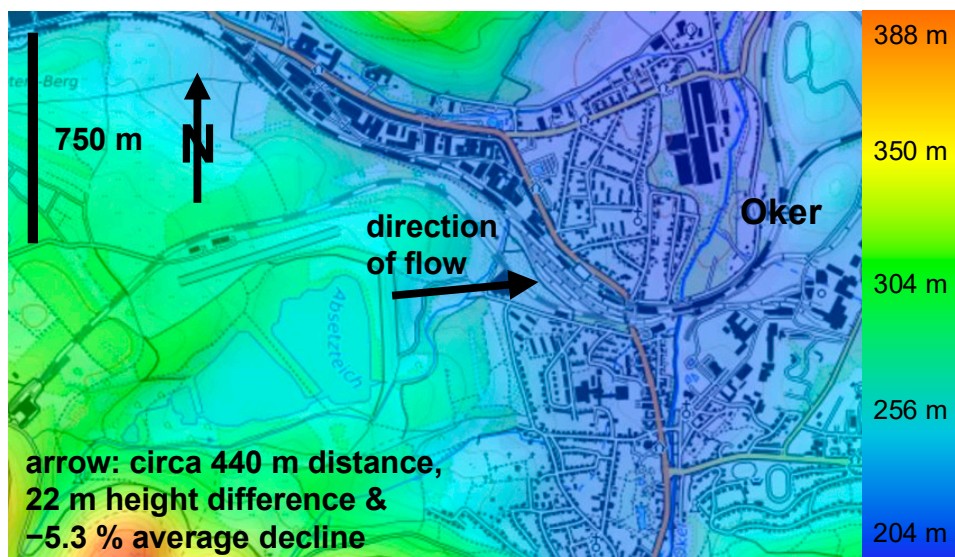

**Figure A1.** Contour map of the area around the TSF Bollrich and assessed direction of tailings flow in case of TSF failure (adapted after topographic-map.com [82]).

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
