# Peer review of "How to Identify Potentials and Barriers of Raw Materials Recovery from Tailings? Part I: A UNFC-Compliant Screening Approach for Site Selection"

_resources, doi:10.3390/resources10030026_

Round 1

Reviewer 1 Report

The readability of the paper could be improved, as it doesn't follow the traditional format, so you need to be clear moving within sections.  Something I would like enhanced would be the conclusions and recommendations.  You have too many bullet points throughout your manuscript and that is a very elementary way to write, please rewrite this section.  Moreover, add more depth and detail about how this information extrapolates in a larger context, e.g. the world.  What can be done to improve or address this?

Reviewer 2 Report

Dear Authors,

The contents of the manuscript represent a coherent, well-reasoned work, and they make a valuable contribution to the existing body of knowledge. The presentation is well arranged and comprehensive. Only some minor issues are identified which should be improved.

  1. The title of the manuscript should be improved. The main (first) part of the title is simply “Anthropogenic Resource Development”, which is not very clear and as such a broad but still unspecific term this is also somewhat misleading for this work, because overall it does not become clear why then the rather very specific topic discussed in the paper would be a “Part 1” of this. Furthermore, “Tailings Mining Project” is also not very precise, because actually the work is about tailing storage facilities. Furthermore, probably it is among the key common characteristics of a screening approach that it is meant to be quick and efficient, and therefore also that expression should be rethought. Overall, the title should be changed to be clearer, and selected so that it better reflects the contents of this work.

  1. Similarly, the list of keywords should be improved, by selecting more specific keywords that better reflect the contents of the work. In particular, the fist keyword (“anthropogenic resources”) is not a convincing choice.

  1. In Section 3, an additional subheading (sub-chapter) should be introduced for the contents presented here in the first part of this section. Accordingly, the numbering of the other sub-chapters must be adapted.

Reviewer 3 Report

The paper was revised according to the journal rules. The main topic treated deserves to be considered for publication. Few revisions are required and reported below:

  • a graphical abstract should be added
  • Hihlights should be considered 
  • Nomenclature list should be added using also si unit of measure 
  • Explain acidic seepage term using AMD as acronym
  • I am not sure that for the citation is necessary to add also pp. Information 

Reviewer 4 Report

In this article, authors proposed a quick and efficient method for screening tailings storage facilities. The main purpose is to evaluate the potentials and barriers of tailings projects. By using this method, a TSF inventory of tailings storage facilities that meets the criteria to be assessed in a more advanced study was created, which can effectively solve the problem of shortage of mineral resources. However, I thought this work has some deficiencies and I recommend to revision before acceptable publication. Detailed comments are listed below:

  1. There are too many keywords. I suggest reducing some words.
  2. In the introduction, it is mentioned that tailings storage facilities have many hazards, but there are many methods or means to reduce these hazards at present. I think this is also important for this work. Please give a brief introduction to these methods or means, You could find some usefull suggestions in papers (Some developments and new insights for environmental sustainability and disaster control of tailings dam, Journal of Cleaner Production,269(2020). Effect of substitution reaction with tin chloride in thermal treatment of mercury contaminated tailings, Environmental Pollution,264(2020). Some developments and new insights of environmental problems and deep mining strategy for cleaner production in mines, Journal of Cleaner Production,210(2019), 1562-1578.).
  3. The citation format used in the article is inconsistent(such as [e.g. 29,30,31], [16-21], and [cf. 28]), please unify.
  4. Has there been any previous study on screening tailings storage facilities? If so, please write in the introduction.
  5. In section 3, What is the basis of “the 5 steps”? Among the five steps, can the order of step (2), step (3), and step (4) be changed?
  6. In table 3, 7 precondition factors are not enough? Geological conditions should also be considered.
  7. In table 4, the consideration of “Category” is not comprehensive enough. I think “air” is also an important environmental factor and should be considered.
  8. In section 3.3, is “(cf. Table 8)” correct in the sentence “The criteria seismic hazard, aqueduct water risk, Fragile States Index and human footprint are adopted (cf. Table 8)”?
  9. Please check the format of the references, use italics or not.

Round 2

Reviewer 1 Report

Thank you for addressing my comments.

Reviewer 4 Report

The authors have reasonably explained and appropriately modified all issues. The manuscript is acceptable.